microbiology/health and disease and epidemiology/biotechnology

honeybee, American foulbrood disease, chalkbrood disease, agrochemical exposure, bacterial and fungal compositions

**Author for correspondence:**
Bin Zhou
e-mail: bzhou@yzu.edu.cn

# Microbiota dysbiosis in honeybee (*Apis mellifera* L.) larvae infected with brood diseases and foraging bees exposed to agrochemicals

Man-Hong Ye[1], Shu-Hang Fan[1], Xiao-Yuan Li[1], Islam Mohd Tarequl[1], Chun-Xiang Yan[4], Wan-Hong Wei[2], Sheng-Mei Yang[2] and Bin Zhou[3]

[1]College of Bioscience and Biotechnology, [2]Joint International Research Laboratory of Agricultural & Agri-Product Safety, and [3]College of Animal Science and Technology, Yangzhou University, Yangzhou 225009, Jiangsu Province, People's Republic of China
[4]Chunxiang Professional Beekeeping Cooperatives, Yangzhou 225009, Jiangsu Province, People's Republic of China

M-HY, 0000-0002-1969-4810; BZ, 0000-0002-0824-964X

American foulbrood (AFB) disease and chalkbrood disease (CBD) are important bacterial and fungal diseases, respectively, that affect honeybee broods. Exposure to agrochemicals is an abiotic stressor that potentially weakens honeybee colonies. Gut microflora alterations in adult honeybees associated with these biotic and abiotic factors have been investigated. However, microbial compositions in AFB- and CBD-infected larvae and the profile of whole-body microbiota in foraging bees exposed to agrochemicals have not been fully studied. In this study, bacterial and fungal communities in healthy and diseased (AFB/CBD) honeybee larvae were characterized by amplicon sequencing of bacterial 16S rRNA gene and fungal internal transcribed spacer1 region, respectively. The bacterial and fungal communities in disordered foraging bees poisoned by agrochemicals were analysed. Our results revealed that healthy larvae were significantly enriched in bacterial genera *Lactobacillus* and *Stenotrophomonas* and the fungal genera *Alternaria* and *Aspergillus*. The enrichment of these microorganisms, which had antagonistic activities against the etiologic agents for AFB and CBD, respectively, may protect larvae from potential infection. In disordered foraging bees, the relative abundance of bacterial genus *Gilliamella* and fungal species *Cystofilobasidium macerans* were significantly reduced, which may compromise hosts' capacities in nutrient absorption and immune defence against pathogens. Significantly higher frequency of environmentally

derived fungi was observed in disordered foraging bees, which reflected the perturbed microbiota communities of hosts. Results from PICRUSt and FUNGuild analyses revealed significant differences in gene clusters of bacterial communities and fungal function profiles. Overall, results of this study provide references for the composition and function of microbial communities in AFB- and CBD-infected honeybee larvae and foraging bees exposed to agrochemicals.

# 1. Introduction

Honeybee (*Apis mellifera* L.), one of the crop pollinators of great socioeconomic importance, is susceptible to infection by a variety of organisms (including bacteria, fungi, viruses and parasites). Of all the diseases, American foulbrood (AFB) disease [1] and chalkbrood disease (CBD) [2] are two infectious diseases that affect honeybee broods and cause substantial economic losses to beekeepers. The spore-forming bacterium *Paenibacillus larvae* and the fungus *Ascosphaera apis* are the causative agents for AFB and CBD, respectively. The persistent nature of their spores, combined with several honeybee behaviours (such as honey robbing and adult workers drift) [3,4], management activities (such as the migration of apiaries, trading of beehive products, reuse and exchange of wax combs across colonies) and higher within-colony densities in the foraging season [5] had increased both between-colony and between-apiary transmission of these pathogens.

Besides biotic stressors, abiotic factors such as agrochemicals (including insecticides, fungicides and herbicides) [6] and residues of in-hive antibiotics [7] can also reduce the fitness of honeybees and increase colony failure [8].

It has been increasingly recognized that honeybee gut microbiota is involved in hosts' metabolism, development and immunity [9]. The relatively constant intestinal microflora plays an important role in maintaining honeybee health [10]. In a recent study, the profiling of gut and whole-body microbiota was suggested to be used to distinguish thriving and non-thriving honeybee hives [11]. However, various factors (such as nutritional deficiencies, pathogens, pesticides and environmental pollution) can cause the perturbation of microbiota. This may increase hosts' susceptibility to pathogens, weaken their abilities in immunomodulation and compromise their health [12]. For migratory apiaries, the microbiota of honeybees are more inclined to be influenced by exposure to the new environment (novel microbes and habitats, changed climate, potential agrochemicals etc.), which may sensitize the colony to various stressors, destabilize the microflora and trigger the outbreak of diseases in potentially unhealthy colonies.

Gut microflora alterations associated with biotic and abiotic stressors have been investigated in honeybees. Bacterial communities in workers and pupae from AFB-affected honeybee colonies have been studied [13]. Transcriptomic investigation on larval gut infected by *A. apis* has been performed in *Apis cerana cerana* [14]. Recently, the number of culturable aerobic gut bacteria in nurse bees from healthy and CBD-infected honeybee colonies was compared. The decrease of this number was suggested as a prognostic marker for the outbreak of CBD [15]. In the meantime, accumulating evidence suggests that pesticides and herbicides, such as thiacloprid [16], thiamethoxam, fipronil, boscalid [17], nitenpyram [18] and glyphosate [19,20], could significantly affect the compositions of intestinal bacterial communities in worker bees. Up until now, the vast majority of microbiota research has focused on the intestinal bacteriobiota of adult honeybees. To our knowledge, the bacterial and fungal compositions in AFB- and CBD-infected larvae and the profile of whole-body microbiota in foraging bees exposed to agrochemicals have not been fully studied.

In this study, we explored (i) the bacteriobiota and mycobiota in AFB- and CBD-affected larvae, respectively, in two migratory apiaries that reported the sporadic occurrence of AFB and CBD; (ii) the compositional and structural shifts taking place in the microbiome of foraging bees suspected of being exposed to agrochemicals in a third migratory apiary. We aimed to characterize and compare the bacterial and (or) fungal compositions between healthy and diseased larvae, as well as between healthy foraging bees and disordered ones exposed to agrochemicals. Our results may provide insights into the relationship between perturbed microbiota and health status of larvae/honeybees under pathogenic and environmental stressors.

# 2. Material and methods

## 2.1. Sample collection

From 5 to 12 April 2019, close to the ending of the full-blooming period of oilseed rape (*Brassica napus* L.), three migratory apiaries of western honeybee (*Apis mellifera* L.) reported sporadic occurrences

of AFB disease, CBD, and suspected poisoning of agrochemicals, respectively. These three apiaries were located in the villages of Yangzhou city, Jiangsu Province, China, belonged to different beekeepers, and were at least 3.5 km apart from each other. Honeybees were all housed in 10-frame deep Langstroth bee boxes.

In the apiary (migrated from Zhejiang Province, China) suspected of being infected with AFB disease, larvae showing clinical symptoms of AFB were detected in only three brood frames in one hive. During a regular afternoon check of colonies, the experienced beekeeper noticed faint and unfavourable odour emitting from one examined frame. A further examination revealed that the source of the odour was sporadic larvae (aged from 48 to 96 h old) in brood nests which were darker than the normally translucent and lustrous larvae. No other AFB-diseased colonies within the apiary were detected after a thorough visual inspection. The remaining colonies in the same apiary were presumed as healthy. For larvae sampling, six larvae per healthy colony were collected with sterile tweezers and pooled together in plastic tubes. Since only one hive showed the existence of AFB-infected larvae, all diseased larvae were from the same colony. Six diseased larvae were pooled together in one tube as described for the healthy ones. Larvae collected in the diseased and healthy colonies were designated as group AFB and group CT.AFB, respectively. After sample collection, the affected hive together with all equipment potentially related to the diseased colony were burned to limit the spread of pathogens.

In the apiary (migrated from Guizhou Province, China) suspected of being inflicted with CBD, larvae with mummified chalky appearances were scattered in 11 colonies. The proportion of infected colonies was approximately 5.1% (11 out of 216). The diseased larvae had white-coloured masses under the skin. Some were fully covered with a layer of fungal mycelium. Six larvae per diseased/healthy colony in the same apiary were extracted aseptically from brood cells, which were designated as group CBD and group CT.CBD, respectively.

In the apiary (migrated from Jilin Province, China) that reported foraging bees (foragers) being poisoned by agrochemicals, an unusual number of dying foragers were observed in approximately 5% colonies (10 out of 198 colonies). Disordered foragers exhibited symptoms suspected of pesticide poisoning which included unnaturally quick movements on the ground and lack of vitality after their return flight home at approximately 15.45–16.30. Most of them died in close proximity to hives' entrances. For sample collection, foragers were collected at the entrance of the hive when they returned to the hives in the afternoon. Three disordered foragers per colony were collected and pooled, which were designated as group DIS. At the same time, three foragers per colony from colonies exhibiting no abnormal symptoms within the same apiary were collected, which were designated as group CT.DIS. In this study, eight replicates were prepared for each group. Samples were immediately freeze-killed by burying them in dry ice, conveyed to the laboratory and stored at −80°C until DNA extraction.

## 2.2. Amplicon sequencing of the bacterial 16S rRNA gene and fungal internal transcribed spacer region

In order to determine potential changes in microbiota compositions, larvae samples from AFB-infected apiary (group AFB and group CT.AFB) and CBD-infected apiary (group CBD and group CT.CBD) were subjected to amplicon sequencing of the 16S rRNA gene and internal transcribed spacer (ITS)1 region, respectively. Foragers from group DIS and group CT.DIS were subjected to both 16S rRNA gene and ITS1 region amplicons sequencing. Sequencing of this study was performed at Novogene Biological Information Technology Co., Ltd, Beijing, China.

A previously described CTAB/phenol-based extraction protocol [21] was used to extract total genomic DNA (gDNA) from larvae/forager samples. To this end, forager samples were prepared as follows. Individual forager was first rinsed in ample sterile water three times to remove the microorganisms on its body surface. Then, three foragers per colony were pooled together in 50 ml sterile tubes and homogenized with sterilized distilled $H_2O$ (1%, w/v) using a homogenizer. The obtained homogeneous solution was filtered through sterilized one-layer gauze to remove the floating slimy materials and then centrifuged at 3500$g$ for 25 min. The resulting pellets, which contained microbes from the guts and other parts of honeybees, were collected for further DNA extraction.

Primer pairs (515F: 5′-GTGCCAGCMGCCGCGGTAA-3′; 806R: 5′-GGACTACHVGGGTWTCTAAT-3′) and (ITS5-1737F: 5′-GGAAGTAAAAGTCGTAACAAGG-3′; ITS2-2043-R: 5′-GCTGCGTTCTTCATC-GATGC-3′) were used to amplify the hyper-variable V4 region of the 16S rRNA gene of bacteria and the ITS1 region of fungi, respectively. PCR amplifications were conducted in a total reaction volume of 25 µl, using 12.5 µl of Phusion® High-Fidelity PCR Master Mix with GC Buffer (New England Biolabs,

Beijing, China), 2.5 µl of each primer (5 µM) and approximately 10 ng of template gDNA. The procedures for PCR amplification were as follows: an initial denaturation step at 98°C for 2 min, followed by 25 cycles of 98°C for 30 s, 55°C for 30 s and 72°C for 30 s, ended with a final extension step at 72°C for 5 min. Amplicons were visualized by 2% agarose gel electrophoresis and purified using a Thermo GeneJET Gel Extraction Kit (Thermo Fisher Scientific, Shanghai, China). The sequencing library was constructed by using Ion Plus Fragment Library Kit (48 rxns, Thermo Fisher Scientific) according to the manufacturer's instructions. After being quantified using a Qubit 2.0 Fluorometer (Thermo Fisher Scientific) and the assessment of the size on an Agilent Bioanalyzer 2100 system (Agilent Technologies, Santa Clara, CA), libraries were pooled in equimolar amounts and subjected to sequencing on an Ion S5$^{TM}$XL (Thermo Fisher Scientific) platform according to standard protocols.

## 2.3. Bioinformatical analyses of sequences

The brief processing steps were as following: Cutadapt (v. 1.9.1, http://cutadapt.readthedocs.io/en/stable/) was used to filter out reads of low quality using the default parameters. Sequencing data from different samples were de-multiplexed according to the barcode sequences. Raw data were obtained after trimming the barcode and primer sequences. Then, clean reads were acquired after the removal of chimeric sequences (https://github.com/torognes/vsearch/). Clean reads were clustered into operational taxonomic units (OTUs) at an identity threshold of 97% similarity by using the UPARSE software (v. 7.0.1001, http://www.drive5.com/uparse/). The tag sequence with the highest abundance was selected as the representative sequence of each OTU, which was used to perform taxonomic assignment by using the SILVA132 database (https://www.arb-silva.de/) and the fungal Unite database (v. 7.2, https://unite.ut. ee/) to describe the taxonomic bacterial and fungal composition of tested samples, respectively.

Analyses of alpha- and beta-diversity were performed with QIIME (v. 1.9.1) software. Indexes, including the observed species, Chao1 and ACE (describing community richness), the Shannon index, the Simpson index (describing community diversity), the Good's coverage and the PD whole tree index were analysed for alpha-diversity differences between groups from the same apiary. To investigate patterns of microbial community diversity, unweighted and weighted UniFrac distances and Bray–Curtis distance matrices were calculated using QIIME. The resulting dissimilarity matrices were further analysed via non-metric multidimensional scaling (NMDS) to visualize the clustering of samples by using R package 'vegan'. Hypothesis testing was carried out using permutational multivariate analysis of variance (PERMANOVA) ('adonis' function in the vegan package) based on Bray–Curtis dissimilarity distance matrices to determine differences in microbial communities between groups [22].

Biomarker features in each group were screened by Metastats (version 1.0) and LEfSe was used for the quantitative analysis of biomarkers within different groups [23]. The contributions of the microorganisms to the differences between groups were evaluated using the linear discriminant analysis (LDA) score and a LDA score larger than 4 indicated a significantly higher relative abundance in the corresponding group ($p < 0.05$).

PICRUSt (Phylogenetic Investigation of Communities by Reconstruction of Unobserved States) (http://picrust.github.io/picrust) was used for predictive functional genomic analyses of bacterial communities. Bacterial function predictions were categorized into Kyoto Encyclopedia of Genes and Genomes (KEGG) pathways. FUNGuild (https://github.com/UMNFuN/FUNGuild) database was used for fungal functional prediction [24].

## 2.4. Statistical analysis

All statistical analyses were performed using R (v. 2.15.3) software. Student $t$-test and non-parametric Wilcoxon test were performed for comparisons between two groups from the same apiary. All $p$-values were adjusted with the Benjamini and Hochberg method to reduce the false discovery rate.

# 3. Results

## 3.1. General information about sequencing

Our sampling regimen resulted in a total of 62 distinct barcoded amplicons (two samples from group CT.CBD failed to produce specific bands on gel and were excluded from the study). A total of 2 528 446 and 2 251 474 clean reads were obtained from high-throughput sequencing of 16S rRNA and

ITS1 region amplicons, respectively. The number of sequences per sample in each group ranged from 69 577 to 81 754. General information about the sequencing results are listed in electronic supplementary material, file S1.

The number of OTUs detected in larvae from two diseased groups (214 in group AFB and 70 in group CBD) was significantly ($p < 0.05$) lower than that detected in larvae from the corresponding healthy colonies (879 in group CT.AFB and 473 in group CT.CBD). In samples from group DIS, ITS amplicons sequencing yielded an average of 421 OTUs, which was significantly ($p = 0.047$) higher than the number (320) in group CT.DIS. Similarly, results of the 16S rRNA amplicons sequencing showed that more OTUs were detected in group DIS (474) than that in group CT.DIS (358) ($p = 0.342$).

## 3.2. Overall microbiota profile of different groups

Our results revealed significant changes in the taxonomic composition of microbiota between diseased and the presumed healthy groups.

In terms to bacterial communities, the most abundant phylum in larvae from group CT.AFB was Proteobacteria (54.49%), followed by Firmicutes (26.64%) and Bacteroidetes (11.43%). In group AFB, the predominant phylum shifted to Firmicutes (98.43% including 97.75% of the genus *Paenibacillus*), while the relative abundance of Proteobacteria was significantly decreased to 1.28% ($p = 0.002$). When group DIS was compared with group CT.DIS, both groups were dominated by Proteobacteria (54.26% and 67.07%, respectively). A significant alteration was observed in the composition of the phylum Firmicutes with foragers in group DIS had significantly higher relative abundance (34.51%) than that in group CT.DIS (19.04%) ($p = 0.038$).

Concerning the fungal communities, Ascomycota was the most abundant phylum in larvae from group CT.CBD (26.96% including 14.37% of the genus *Ascosphaera*), followed by Basidiomycota (2.83%). In the diseased group CBD, a strong dominance of Ascomycota was observed (99.94% including exclusively the genus *Ascosphaera*). Similar trends of increased relative abundance of Ascomycota and decreased proportion of Basidiomycota were also observed in disordered foragers. Foragers from group DIS had a significantly higher ($p = 0.001$) proportion of Ascomycota (23.57%) when compared with that in group CT.DIS (1.90%), accompanied by a significant reduction ($p = 0.006$) in the abundance of Basidiomycota (from 7.27% in group CT.DIS to 1.70% in group DIS). Community structure with regard to the relative abundance of top 10 taxa at the genus level in different groups is shown in figure 1.

## 3.3. Microbial community differences between groups from the same apiary

Results from the analysis of alpha-diversity (figure 2) revealed significant decreases in the number of observed species and the Shannon index in group AFB and group CBD, which indicated that the bacterial/fungal communities in diseased larvae were significantly less rich and less even than that in healthy larvae.

When group DIS was compared with group CT.DIS, significant differences in alpha-diversity indexes were observed in the fungal community composition. Disordered foragers had significantly higher fungal community richness. Results based on 16S rRNA sequencing showed that alpha-diversity indexes in group DIS, including the observed species, Shannon and Simpson indexes, Chao1, ACE and PD whole tree, all exhibited a trend of statistically non-significant increase in value when compared with group CT.DIS. Our results indicated that disordered foragers may have more opportunistic or transient colonizers. Detailed results of the alpha-diversity analysis are shown in electronic supplementary material, file S2.

The detailed changes in the microbes between different groups from the same apiary were further revealed by LEfSe analysis (figure 3).

At the genus level, group AFB showed a significant enrichment of *Paenibacillus* (*Paenibacillus larvae*) with an LDA score of 5.67 ($p < 0.001$), while group CT.AFB was significantly enriched in *Stenotrophomonas* ($p = 0.0008$), *Bombella* ($p = 0.040$), *Lactobacillus* (*Lactobacillus acetotolerans*, $p = 0002$), unidentified *Lachnospiraceae* ($p = 0006$) and *Gallicola* ($p = 0.016$) with LDA scores of 5.22, 4.50, 4.38, 4.29 and 4.18, respectively. These differences were probably responsible for the reduction in alpha-diversity in group AFB as compared with group CT.AFB. When group DIS and group CT.DIS were compared, Firmicutes was significantly enriched in group DIS ($p = 0.036$, LDA score 4.92), while *Spiroplasma* and *Gilliamella* were significantly enriched in group CT.DIS ($p < 0.030$) with LDA scores of 4.73 and 4.82 respectively.

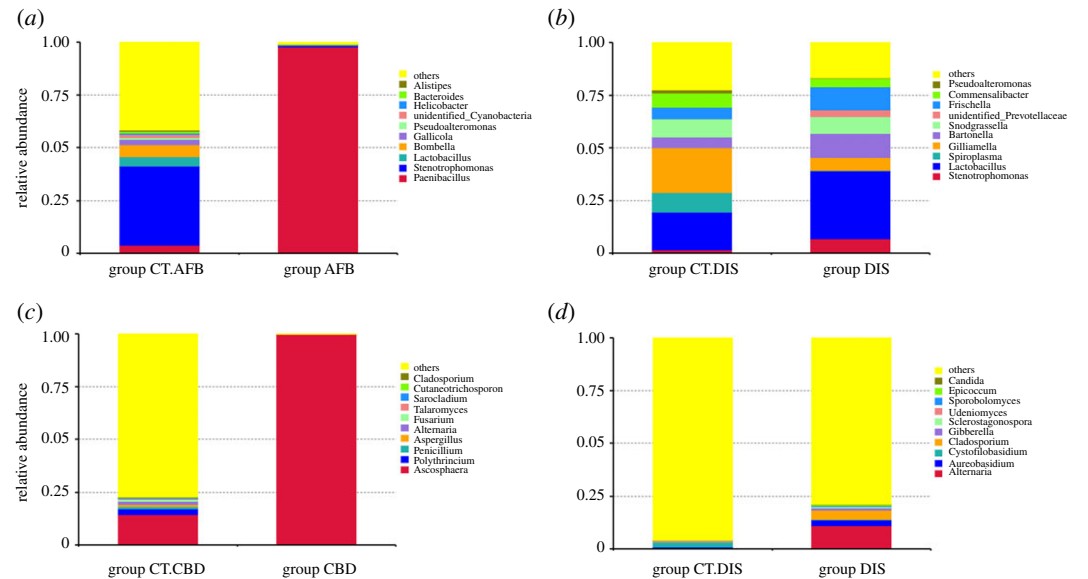

**Figure 1.** Relative abundance of the top 10 taxa at the genus level calculated for each group. (*a,b*) Bacterial microbiota. (*c,d*) Fungal microbiota.

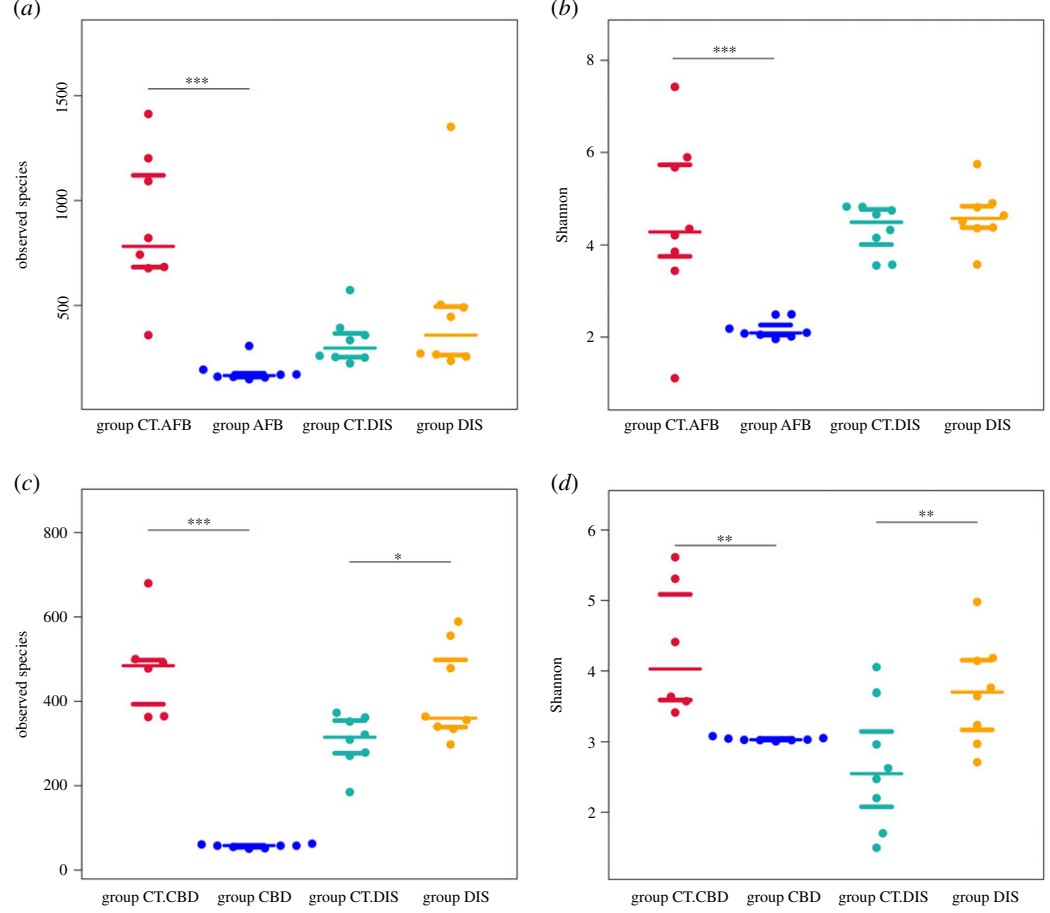

**Figure 2.** Plot of mean alpha-diversity for different groups. Community richness and diversity is characterized by the observed species and the Shannon index, respectively. (*a,b*) Bacterial communities. (*c,d*) Fungal communities. Asterisks indicate a significant difference between groups from the same apiary. $^{*}p < 0.05$, $^{**}p < 0.01$, $^{***}p < 0.001$.

*Ascosphaera* (*Ascosphaera apis*) was selectively enriched in group CBD with an LDA score of 5.57 (*p* = 0002). *Alternaria* (*Alternaria alternata*) and *Polythrincium* (*Polythrincium trifolii*), with an LDA score of 4.13 and 4.19, respectively, were the two genera significantly enriched in group CT.CBD. When

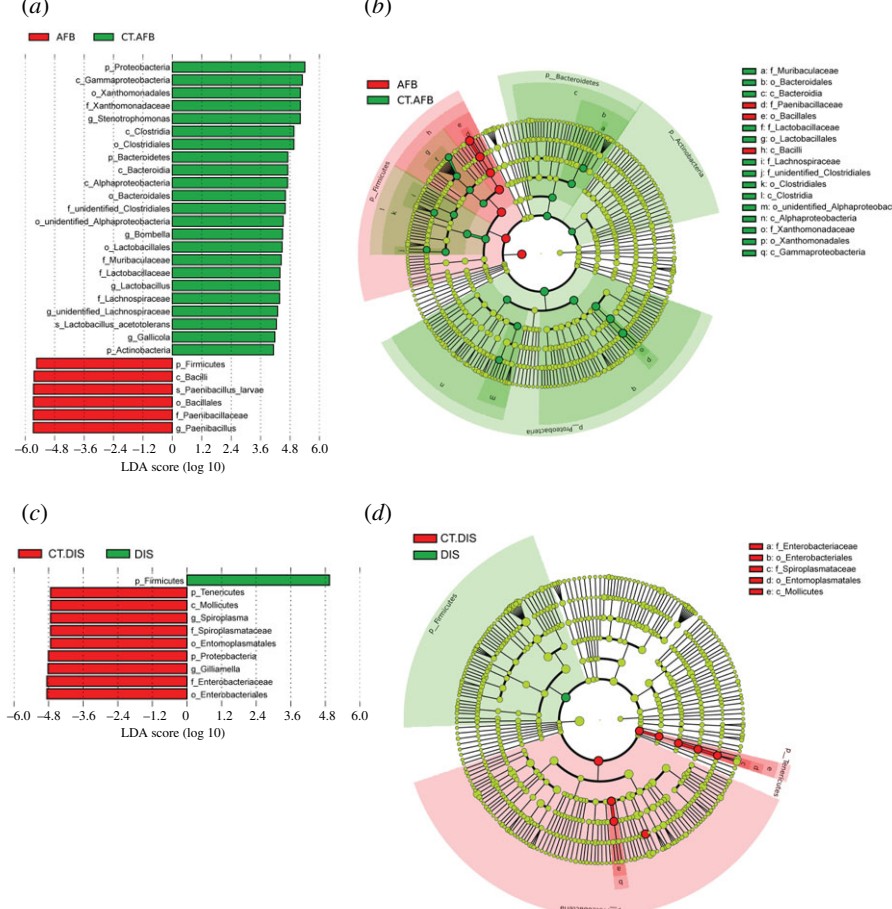

**Figure 3.** Significantly different taxa between groups from the same apiary identified by LEfSe. (a, c, e and g) LefSe analysis (the threshold of the LDA score was 4.0). (b, d, f and h) Taxonomic cladogram based on the results of LEfSe analysis. The size of the circles was based on relative abundance. Yellow represents no significant difference in taxa; green and red represent significantly different taxa in the correspondingly colour-labelled groups. Only taxa with greater than 0.1% proportional abundance are visualized. (a, b, c, d), based on 16S rRNA sequencing data. (e, f, g, h), based on ITS sequencing data.

group DIS and group CT.DIS were compared, LEfSe identified *Alternaria* (*Alternaria alternata*) and *Cladosporium* (*Cladosporium chasmanthicola*), with an LDA score of 5.06 and 4.34, respectively, as the significantly enriched genera in group DIS ($p < 0.021$).

Furthermore, non-metric multidimensional scaling (NMDS) based on Bray–Curtis dissimilarities revealed a clear separation of samples between diseased larvae (group AFB and group CBD) and the corresponding healthy larvae (group CT.AFB and group CT.CBD respectively), indicating that the microbial community compositions between these groups were different from each other (figure 4). Consistently, pairwise PERMANOVA comparisons also indicated that there were significant differences in microbial communities between larvae from two AFB-related groups ($F = 16.805$, $R^2 = 0.54553$, $p = 0.001$) and larvae from the two CBD-related groups ($F = 23.191$, $R^2 = 0.65901$, $p = 0.001$).

Concerning group DIS and group CT.DIS, the NMDS plot showed some separation in fungal communities, but no separation in bacterial communities (overlapped). Results from the PERMANOVA analyses demonstrated that comparisons between foragers in these two groups were significantly different in fungal communities ($F = 2.5537$, $R^2 = 0.15427$, $p = 0.029$) and bacterial communities ($F = 2.3877$, $R^2 = 0.1457$, $p = 0.005$). Taken together, these results suggested that the intra-group differences were much smaller than the inter-group differences. Meanwhile, the relatively smaller $R^2$ value indicated that other sources of variation may exist besides the composition of bacterial and fungal communities.

## 3.4. Predicted microbial function classification via PICRUSt and FUNGuild

Results from the PICRUSt analysis indicated that changes in the bacterial taxa in AFB-diseased larvae altered the microbiota function. The analysis of KEGG functional classes (level 3) revealed significant

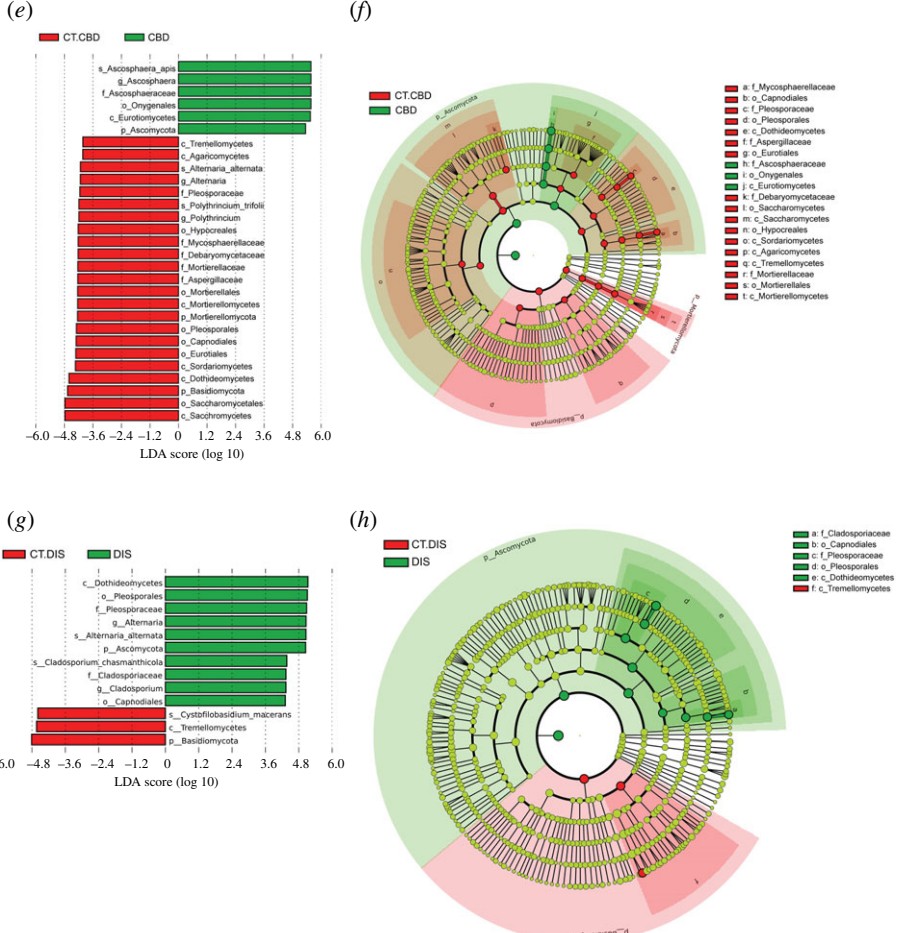

**Figure 3.** (Continued.)

differences in functional gene categories between group AFB and group CT.AFB. The amino sugar and nucleotide sugar metabolism, sporulation and glycolysis/gluconeogenesis gene clusters were significantly more abundant in AFB-diseased larvae. On the other hand, diverse gene clusters, including DNA repair and recombination protein, purine metabolism, ribosome, secretion system, peptidases, oxidative phosphorylation, ribosome biogenesis, chaperones and folding catalysis were more abundant in larvae from group CT.AFB. When group DIS was compared with group CT.DIS, differentially enriched genes included energy metabolism and galactose metabolism-related genes, and genes involved in the biosynthesis of lipopolysaccharide and streptomycin (figure 5a).

Results from the FUNGuild analysis indicated that the relative abundance of the animal pathogen in group CBD was significantly higher than that in group CT.CBD. Similarly, group DIS had significantly more animal and plant pathogen-related function guilds than group CT.DIS. In addition, FUNGuild analysis also suggested that an unassigned function guild was significantly higher in relative abundance in both group CT.CBD and group CT.DIS when compared with their corresponding diseased groups (figure 5b).

# 4. Discussion

In the present study, significant differences in microbial compositions were observed between larvae from healthy and diseased colonies. In healthy larvae, the top 10 bacterial and fungal genera accounted for 58.29% and 22.92% of the whole microbial composition in group CT.AFB and group CT.CBD respectively. The rest was mainly composed of low-abundance (relative abundance less than 0.1%) microorganisms, which were actually the main part that contributed to the community diversity. In infected larvae, the pathogenic *P. larvae* and *A. apis* comprised the overwhelming majority of the bacterial and fungal microbiota in larvae from group AFB (96.48%) and group CBD

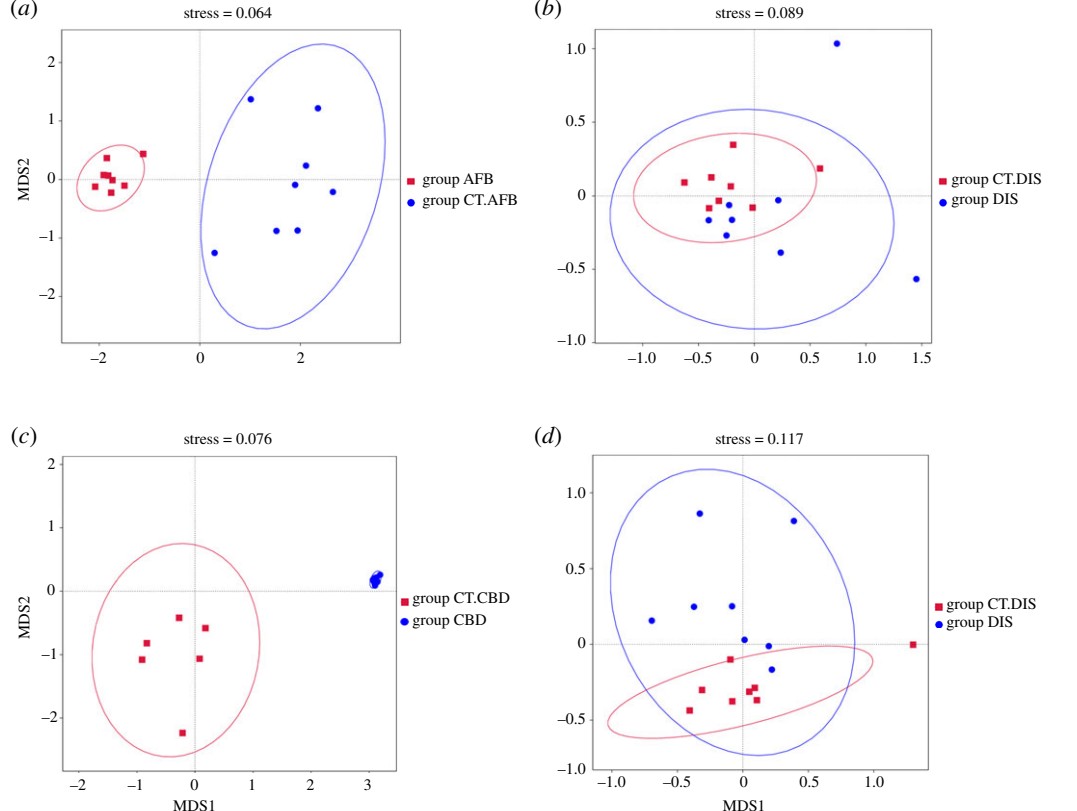

**Figure 4.** Non-metric multidimensional scaling (NMDS) plots showing the bacterial (a,b) and fungal (c,d) community diversity between groups from the same apiary.

(99.89%) respectively. The abundance of other microorganisms was too low to be detected. As a result, the community richness and diversity were significantly reduced in diseased larvae.

In the meantime, our results also demonstrated the presence of *P. larvae* (3.75%) and *A. apis* (14.44%) in the presumed healthy larvae from group CT.AFB and group CT.CBD respectively. A detailed look into the data revealed that the relative abundance of *P. larvae* in one sample in group CT.AFB was as high as 23.28%, while this proportion in the other seven larvae samples in group CT.AFB averaged only 0.96%. Similarly, the frequency of *A. apis* detected in group CT.CBD was mainly contributed by three larvae samples (averaged 28.17%), while the other three samples averaged only 0.71%. Our present results indicated that *P. larvae* and *A. apis* may persist as asymptomatic infections in these presumed healthy colonies.

In fact, detection of *P. larvae* in brood and adult honeybee samples in asymptomatic colonies was not uncommon [25–27]. *P. larvae* may exist as a pathobiont in the microbiota of worker bees, from where it is transmitted to broods [13]. Colonies infected by *P. larvae* but without manifested clinical symptoms could be attributed to variations in hosts' tolerance, pathogens' virulence, the density of colonies [28] and other abiotic or even random factors [29]. Similarly, *A. apis* could be detected in hives showing no symptoms of infection [30] and resided in hives asymptomatically as inactive spores.

Data from high throughput sequencing of 16S rRNA gene and ITS1 region amplicons enabled us to analyse the dynamic structure of microbial composition in larvae under different physiological conditions. The bacterial communities observed in honeybee larvae (second to fourth instars) in this study were consistent with previous results which demonstrated that honeybee larvae were dominated by Acetobacteraceae Alpha 2.2 (first and second larval instars) and *Lactobacillus* (later instars) [31]. Our results showed that larvae in group CT.AFB had a significantly higher proportional abundance of *Lactobacillus* (*Lactobacillus acetotolerans*), Lactobacillaceae (family) and Lactobacillales (order) than AFB-diseased larvae. As is known, *Lactobacillus*, an important genus within lactic acid bacteria (LAB), is beneficial to its hosts through the production of antimicrobial metabolites and peptides, as well as through the modulation of immune response [32]. Accumulated results have demonstrated that *Lactobacillus* isolated from larvae or the gut of adult *A. mellifera*, exhibited antagonistic activity against the growth of *P. larvae* [33]. Results from both *in vitro* and *in vivo* studies also demonstrated that *Lactobacillus* could stimulate the innate immune response in honeybees [34]

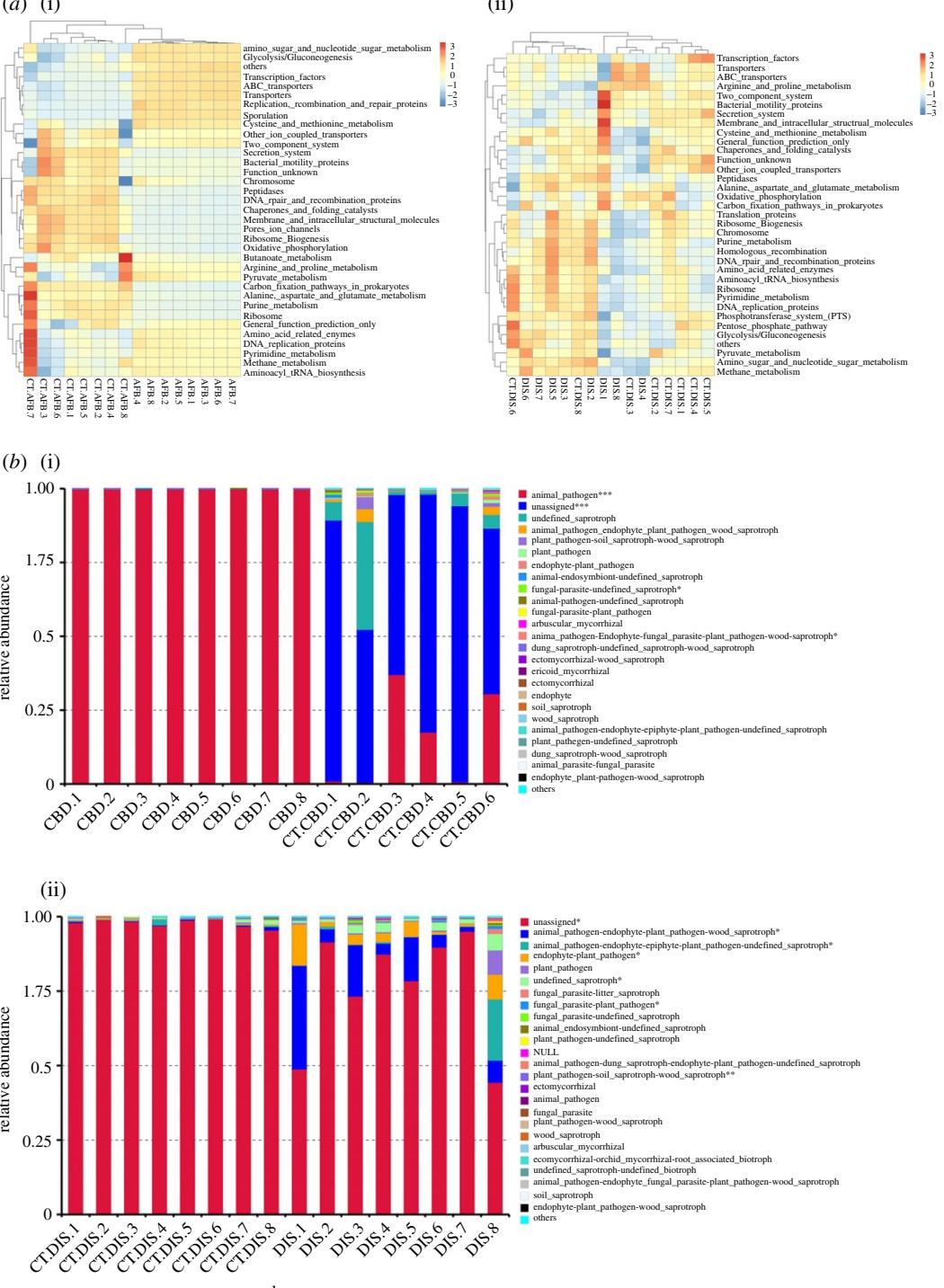

**Figure 5.** (a) Heatmap of microbial function pathways in different groups. (b) Fungal function classification by FUNGuild tool. Asterisks represent significant differences between groups. $^*p < 0.05$, $^{**}p < 0.01$, $^{***}p < 0.001$.

and inhibit the growth of *P. larvae* [35,36]. A recent study suggested the usage of *Lactobacillus* as a probiotic approach to reduce pathogen load and improve honeybee survival towards *P. larvae* infection [37]. Other genera that existed in significantly higher abundance in larvae from group CT.AFB included *Stenotrophomonas* and *Bombella*. The former was reported to present antagonism with *P. larvae* [38]. The latter, belonging to a clade of acetic acid bacteria (AAB) within the family Acetobacteraceae Alpha 2.2, could increase larval fitness [39] in such aspects as carbohydrate utilization, host–microbe interaction and resistance to pathogens [40,41]. Based on these results, we proposed that the presence of much more abundant *Lactobacillus*, *Stenotrophomonas* and *Bombella* in

larvae from group CT.AFB could protect the larvae from succumbing to AFB infection, which may partly explain the asymptomatic status in the presence of *P. larvae*.

Concerning the fungal composition in larvae from group CBD and group CT.CBD, the relative abundance of genus *Alternaria* (*Alternaria alternata*) and family Aspergillaceae in larvae from group CT.CBD were significantly higher than that in group CBD. Genus *Aspergillus* was also present in much higher abundance (not of statistical significance) in group CT.CBD compared with that in group CBD. Recent research demonstrated that *Alternaria* and *Aspergillus* had antagonistic activities against CBD [42]. Their abundant presence in larvae from group CT.CBD may provide larvae with potential protection against CBD infection.

In the present study, the predominance of *A. apis* in larvae from group CBD was expected since larvae sampled in this group showed clinical symptoms of CBD. However, most 'unhealthy' larvae in group AFB didn't have obvious symptoms. Our results indicated that the growth of other bacteria in larvae had been significantly inhibited by *P. larvae* before the initiation of typical symptoms of AFB disease, which was most likely due to the capacities of *P. larvae* to produce various secondary metabolites (including siderophores, paenilarvins, paenilamicins, sevadincin and bacteriocin) to enhance its ability to kill potential microbial competitors [43,44]. Furthermore, the significant decline of non-pathogenic bacteria, which were capable of stimulating the innate immune response of honeybee larvae [45], made larvae more vulnerable to potential pathogens.

It should be noted that testing positive for the presence of *P. larvae* and *A. apis* does not necessarily imply the final outbreak of the disease, which actually depends on the genetic background (susceptibility) [46] and general health status (co-infection with other pathogens) of the honeybees [47], the number and virulence of contaminated pathogens (or spores) [48], and environmental conditions [49]. A variety of stressors, including malnutrition, the presence of parasites, exposure to pesticides and disturbed gut microbiome [50], might increase larvae's susceptibility to pathogens and trigger disease emergence.

In this study, three migratory apiaries housed their colonies in a region with oilseed rape being the principal flora from the beginning of March to the middle of April. We proposed that the lack of varied floral resources throughout this season, together with the reduction of flowers (the oilseed rape flowers started fading) might enhance these infectious risks and lower the threshold for the occurrence of diseases in potentially unhealthy colonies. The other possibility was that honeybees in these migratory apiaries had not been treated with any medicinal treatments for more than one month, which might lead to the appearance of symptoms originally masked by prophylactic antibiotic treatment. Under such circumstances, improved management and sanitation practices were of great importance to keep these diseases in check [51] and to achieve localized extinction of the disease [52]. In fact, the two AFB- and CBD-infected apiaries stopped honey production immediately after the detection of diseased larvae. They focused on such measures as keeping hives clean and well ventilated, enhancing nutritional supplements and avoiding chemical exposure, which proved to be successful to prevent outbreaks of the diseases and future losses stemming from weakened colonies.

When group DIS was compared with group CT.DIS, variations in the relative abundance of certain bacterial and fungal taxa that made up the microbiota of foragers were observed.

Among the bacterial community members harboured in the gut of adult honeybees, six taxa (*Gilliamella*, *Lactobacillus*, *Snodgrassella*, *Bartonella*, *Frischella* and *Bifidobacterium*) were considered to be the core members represented in high proportions [53]. These bacteria were the major gut symbionts and crucial to the health of honeybees with regard to their beneficial effects on nutrient acquisition, pathogen defence and immunity [54]. In the present study, six core gut taxa were all identified in foragers from both group CT.DIS and group DIS, which altogether, accounted for 62.03% and 75.01% of the whole bacterial communities, respectively. In previous reports, the proportion of these core bacterial members in the gut of adult honeybees accounted for over 95% of the community [55,56]. We attributed this difference to different sampling methods. In this study, the whole-body microbiota of foragers was analysed, not only the intestinal flora, but also the microorganisms in other parts of honeybees (such as the mouth parts and the hypopharyngeal glands), the inclusion of which may reduce the proportion of the top 10 microbes.

Our results showed that none of the core bacterial members was eliminated from colonization due to the suspected toxicity of agrochemicals. Except for the genus *Snodgrassella*, which remained stable in foragers (8.59% and 8.04% in group CT.DIS and group DIS, respectively), these beneficial bacteria were different in their proportional abundance between foragers from different groups. Our results showed that the relative abundance of *Gilliamella* experienced a significant reduction ($p = 0.018$) in

disordered foragers (21.48% and 6.28% for group CT.DIS and group DIS, respectively). Previous research demonstrated that *Gilliamella* contributed to pectin degradation and the breakdown of pollen walls and was involved in biofilm formation in the gut of honeybees [57]. *Gilliamella apicola*, a dominant gut bacterium in honeybees exclusively catabolizing carbohydrates [58], was capable of using sugars harmful to honeybees and breaking down other potential toxic carbohydrates [59]. The significant decrease in the relative abundance of *Gilliamella* in disordered foragers may compromise hosts' dietary tolerances, weaken hosts' capacities in absorbing nutrients from pollen and sustaining immunocompetence against the invasion of opportunistic pathogens.

Besides *Gilliamella*, genus *Spiroplasma* (*Spiroplasma montanense*) also existed in significantly higher proportional abundance in foragers from group CT.DIS. Among the eight forager samples in group CT.DIS, the relative abundance of *Spiroplasma montanense* (*S. montanense*) in two samples averaged 35.40%, while the other six samples averaged only 0.57%. Given the close association of *spiroplasmas* with flowers [60], together with the great variation in the frequency of *S. montanense* in foragers within group CT.DIS, we proposed that *S. montanense* carried by foragers were more likely to be from forage sources (i.e. nectar and pollen) than from the gut. On the other hand, *S. montanense* was not the etiologic pathogens (*Spiroplasma melliferum* and *Spiroplasma apis*) that caused spiroplasmosis (also referred as 'May disease') in honeybees [61]. Thus, the disordered foragers sampled in our study were less likely to have suffered from spiroplasma infection despite exhibiting similar symptoms of crawling bees on the ground in front of the hive as honeybees infected with spiroplasmosis [62].

As for the proportion of *Lactobacillus*, *Bartonella*, *Frischella* and *Bifidobacterium*, they all showed a trend of increase in disordered foragers, changed from 17.95%, 6.29%, 6.02% and 1.7% in group CT.DIS to 32.29%, 11.70%, 12.18% and 4.53%, respectively, in group DIS. However, due to the greater variability among forager samples, these differences were of no statistical significance. Taken together, the lack of significantly enriched bacterial pathogens, together with the non-significantly increased proportion of total beneficial bacteria in foragers from group DIS, confirmed our conclusion that the observed disorder in foragers were unlikely driven by the prevalence of pathogenic bacteria.

As for the mycobiota (fungal community) in foragers from group DIS and group CT.DIS, our current results demonstrated that foragers in both groups were dominated by the fungal phylum Ascomycota, followed by Basidiomycetes, which was consistent with the previous research [63,64]. Furthermore, foragers from group CT.DIS were significantly higher in the relative abundance of species *Cystofilobasidium macerans* than foragers from group DIS. Previous reports demonstrated that strains from genus *Cystofilobasidium* were producers of pectinase, including polygalacturonase [65] and pectin lyase [66], which were important for the degradation of the polysaccharide walls of pollen grains. *Cystofilobasidium macerans* was also capable of secreting diverse extracellular enzymes and exhibited cellulytic and proteolytic [67] activities besides the pectinolytic activity. The significant reduction in the relative abundance of *C. macerans* in disordered foragers may impair their capacities in nutrient utilization.

On the other hand, a significantly higher frequency of *Alternaria* (*Alternaria alternata*) and *Cladosporium* (*Cladosporium chasmanthicola*) was observed in disordered foragers. *Alternaria alternata* was one of the most common phytopathogens capable of producing phytotoxins, mycotoxins and other secondary toxic metabolites [68,69]. *Cladosporium* was commonly found in pollen [70] and existed in a beehive environment [71]. The existence of these environmentally derived fungi might reflect the features of the foraging sites. It was worthy to note that disordered foragers exhibited greater taxonomic diversity (especially more diverse mycobiota) compared with foragers in group CT.DIS, which reflected the perturbed microbiota communities and imbalanced interactions between them.

Our results indicated that the microbiota changes in foragers from group DIS were more likely caused by transient floral and airborne microbes to which foragers were exposed in the local environment. In the current study, we correlated the unusual death of foragers with agrochemical exposure based on the following facts: (i) no history of viral disease outbreak in the apiary, (ii) no prevalence of pathogenic bacteria or fungi detected, and (iii) death of foragers stopped when beekeepers transferred the apiary to another location 13 km away. However, we do not know whether the alteration of microbiota happened first which led to the increased sensitivity of foragers to potential agrochemicals, or the exposure to agrochemicals resulted in the disturbances of microbiota communities in foragers. In either way, we are still unable to associate their presence with the neurological symptoms observed in disordered foragers before further investigations are performed.

All in all, the current study delineated the alterations of bacterial and fungal communities in honeybee larvae under the stress of AFB and CBD, as well as in foragers poisoned by agrochemicals. Our results demonstrated that the composition of microbiota is closely related to the health status of their hosts.

Ethics. Samples used in this study were collected from the migratory apiaries. No special ethical statement is required for the present study.

Data accessibility. Raw sequencing data obtained in this study are available on the NCBI Sequence Read Archive (SRA). The bacterial sequencing data for larvae and foragers can be accessed through accession numbers PRJN661982 (https://dataview.ncbi.nlm.nih.gov/object/PRJNA661982) and PRJN661983 (https://dataview.ncbi.nlm.nih.gov/object/PRJNA661983) respectively. The fungal sequencing data for larvae and foragers were deposited under the accession numbers PRJN662004 (https://dataview.ncbi.nlm.nih.gov/object/PRJNA662004) and PRJN662019 (https://dataview.ncbi.nlm.nih.gov/object/PRJNA662019) respectively.

Authors' contributions. M.-H.Y., S.-H.F., X.-Y.L. and I.M.T. performed bioinformatics analysis and statistical analysis; M.-H.Y., C.-X.Y. and B.Z. collected samples used in this study; M.-H.Y. drafted the first version of the manuscript; W.-H.W., S.-M.Y. and B.Z. coordinated the study and revised the manuscript. All authors approved the final version of the manuscript prior to submission and agreed to be held accountable for the work performed therein.

Competing interests. We declare we have no competing interests.

Funding. This work was supported by a programme of Yangzhou Science and Technology Bureau (Modern Agriculture) (grant no. SNY2018000031), Screening of Plant-originated Antibiotic Substitutes Used in Honey Bee Colonies.

Acknowledgement. We would like to thank Mr. Ryan Dimmock, who is from School of Pharmacy and Bioengineering, Keele University in the United Kingdom for his proofreading the whole manuscript.

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
