## [Reviewer comments · Royal Society Open Science]

Review History

RSOS-201805.R0 (Original submission)

Review form: Reviewer 1

Is the manuscript scientifically sound in its present form?

No

Are the interpretations and conclusions justified by the results?

Yes

Is the language acceptable?

No

Do you have any ethical concerns with this paper?

No

Have you any concerns about statistical analyses in this paper?

No

Recommendation?

Accept with minor revision (please list in comments)

Comments to the Author(s)

English has to be corrected by a professional proofreader or a holder of CPE certificate.

Review form: Reviewer 2

Is the manuscript scientifically sound in its present form?

Yes

Are the interpretations and conclusions justified by the results?

Yes

Is the language acceptable?

Yes

Do you have any ethical concerns with this paper?

No

Have you any concerns about statistical analyses in this paper?

No

Recommendation?

Accept with minor revision (please list in comments)

Comments to the Author(s)

This is an interesting piece of work that takes advantage of field-realistic conditions to investigate how microbial infections in honey bee larvae or agrochemical poisoning in honey bee foragers can affect microbial communities of honey bees. For that, they used next-generation sequencing to assess the bacterial and/or fungal profiles of sampled larvae/foragers in hives exhibiting signs of disease or agrochemical poisoning, and reported the changes in microbial communities in terms of relative abundance.

I really missed information regarding absolute abundance of the bacterial and fungal communities in those samples, which could have been achieved by performing quantitative PCR. Adding this information to the study would not only improve the confidence of the results (since changes in microbial relative abundance do not always correlate with changes in absolute abundance) but also increase the impact of the study. I strongly recommend performing qPCR assays to investigate the overall bacterial and fungal loads in bee larvae and foragers.

Another thing that caught my attention: more information should be provided regarding the potential agrochemicals used in the area, maybe the sampled bees could be assayed to detect the presence of potential agrochemicals. It is really vague to say that those bees were poisoned without proving that. As the authors mention, many other environmental stressors could be responsible for the observed side effects.

Other comments:

Page 8, Line 25: Briefly explain how foragers were detected and collected. For the affected hives, were the sampled foragers exhibiting signs of poisoning? Were they collected at the entrance of the hive, leaving or returning to the hives?

Page 10, Line 48: Although the OTU method used in the study is completely valid, the new ESV/ASV methods are better in my opinion and should be considered for future work.

Figure 1 – The high proportion of “Others” is at least intriguing and deserves discussion.

Increase font size for Figures 1 and 3. It's really hard to read them.

Page 3, Line 51: Remove the expression: “In the mean time”

Page 4, Line 38: Replace “agent” with “agents”

Page 5, Line 25: Correct the expression: “are more tend to be”

Page 6, Line 27: Replace “microbiom” with “microbiome”

Page 13, Line 35: “Similarly” followed by “However” in Line 40 is controversial

Page 27, Line 49: Replace “recourses” with “resources”

Decision letter (RSOS-201805.R0)

Dear Professor Zhou

On behalf of the Editors, we are pleased to inform you that your Manuscript RSOS-201805 "Microbiota dysbiosis in honey bee (*Apis mellifera* L.) larvae infected with brood diseases and foraging bees exposed to agrochemicals" has been accepted for publication in Royal Society Open Science subject to minor revision in accordance with the referees' reports. Please find the referees' comments along with any feedback from the Editors below my signature.

Please submit your revised manuscript and required files (see below) no later than 7 days from today's (ie 04-Dec-2020) date. Note: the ScholarOne system will 'lock' if submission of the revision is attempted 7 or more days after the deadline. If you do not think you will be able to meet this deadline please contact the editorial office immediately.

on behalf of Dr Ulas Tezel (Associate Editor) and Pete Smith (Subject Editor)
openscience@royalsociety.org

Subject Editor (Prof Pete Smith):

Comments to the Author:

Both reviewers of the manuscript agreed on accepting the manuscript with minor revisions. The first Reviewer suggested only proofreading to improve the language of the manuscript. The second reviewer's comment: "I strongly recommend performing qPCR assays to investigate the overall bacterial and fungal loads in bee larvae and foragers." would require more than a minor revision, so you should treat this as an optional task for the manuscript.

Reviewer comments to Author:

Reviewer: 1

Comments to the Author(s)

English has to be corrected by a professional proofreader or a holder of CPE certificate.

Reviewer: 2

Comments to the Author(s)

This is an interesting piece of work that takes advantage of field-realistic conditions to investigate how microbial infections in honey bee larvae or agrochemical poisoning in honey bee foragers can affect microbial communities of honey bees. For that, they used next-generation sequencing to assess the bacterial and/or fungal profiles of sampled larvae/foragers in hives exhibiting signs of disease or agrochemical poisoning, and reported the changes in microbial communities in terms of relative abundance.

I really missed information regarding absolute abundance of the bacterial and fungal communities in those samples, which could have been achieved by performing quantitative PCR. Adding this information to the study would not only improve the confidence of the results (since changes in microbial relative abundance do not always correlate with changes in absolute abundance) but also increase the impact of the study. I strongly recommend performing qPCR assays to investigate the overall bacterial and fungal loads in bee larvae and foragers.

Another thing that caught my attention: more information should be provided regarding the potential agrochemicals used in the area, maybe the sampled bees could be assayed to detect the presence of potential agrochemicals. It is really vague to say that those bees were poisoned without proving that. As the authors mention, many other environmental stressors could be responsible for the observed side effects.

Other comments:

Page 8, Line 25: Briefly explain how foragers were detected and collected. For the affected hives, were the sampled foragers exhibiting signs of poisoning? Were they collected at the entrance of the hive, leaving or returning to the hives?

Page 10, Line 48: Although the OTU method used in the study is completely valid, the new ESV/ASV methods are better in my opinion and should be considered for future work.

Figure 1 – The high proportion of “Others” is at least intriguing and deserves discussion.

Increase font size for Figures 1 and 3. It's really hard to read them.

Page 3, Line 51: Remove the expression: “In the mean time”

Page 4, Line 38: Replace “agent” with “agents”

Page 5, Line 25: Correct the expression: “are more tend to be”

Page 6, Line 27: Replace “microbiom” with “microbiome”

Page 13, Line 35: “Similarly” followed by “However” in Line 40 is controversial

Page 27, Line 49: Replace “recourses” with “resources”

===PREPARING YOUR MANUSCRIPT===

===PREPARING YOUR REVISION IN SCHOLARONE===

Author's Response to Decision Letter for (RSOS-201805.R0)

See Appendix A.

Decision letter (RSOS-201805.R1)

Dear Professor Zhou,

It is a pleasure to accept your manuscript entitled "Microbiota dysbiosis in honey bee (*Apis mellifera* L.) larvae infected with brood diseases and foraging bees exposed to agrochemicals" in its current form for publication in Royal Society Open Science.

You can expect to receive a proof of your article in the near future. Please contact the editorial office (openscience@royalsociety.org) and the production office (openscience_proofs@royalsociety.org) to let us know if you are likely to be away from e-mail contact – if you are going to be away, please nominate a co-author (if available) to manage the proofing process, and ensure they are copied into your email to the journal.

on behalf of Dr Ulas Tezel (Associate Editor) and Pete Smith (Subject Editor)
openscience@royalsociety.org

Appendix A

Dear Anita and editors,

We would like to thank two referees and two editors profusely for their valuable advice. The point-to-point response to referees and editors are listed below.

Subject Editor (Prof Pete Smith):

Comments to the Author:

Both reviewers of the manuscript agreed on accepting the manuscript with minor revisions. The first reviewer suggested only proofreading to improve the language of the manuscript. The second reviewer's comment: "I strongly recommend performing qPCR assays to investigate the overall bacterial and fungal loads in bee larvae and foragers." would require more than a minor revision, so you should treat this as an optional task for the manuscript.

Reviewer comments to Author:

Reviewer: 1

Comments to the Author(s)

Q: English has to be corrected by a professional proofreader or a holder of CPE certificate.

A: We have invited a native speaker, Ryan Dimmock, who is from School of Pharmacy and Bioengineering, Keele University in the United Kingdom to proofread the whole manuscript for improvement in grammatical/English language. Here attaches his confirmation.

Ryan Dimmock
Keele University
Guy Hilton Research Centre
Thornburrow Drive
ST4 7QB
Email: r.l.dimmock@keele.ac.uk

Wednesday 9th December

Confirmation of Proof Reading

To whom it may concern,

This letter is to confirm that I have both proof-read and suggested grammatical / English language changes to the manuscript "Microbiota dysbiosis in honey bee (*Apis mellifera* L.) larvae infected with brood diseases and foraging bees exposed to agrochemicals" by Man-Hong Ye et al., as a native English speaker.

Kind regards,

Ryan Dimmock
PhD Student in Cell and Tissue Engineering
School of Pharmacy and Bioengineering

Reviewer: 2

Comments to the Author(s)

This is an interesting piece of work that takes advantage of field-realistic conditions to investigate how microbial infections in honey bee larvae or agrochemical poisoning in honey bee foragers can affect microbial communities of honey bees. For that, they used next-generation sequencing to assess the bacterial and/or fungal profiles of sampled larvae/foragers in hives exhibiting signs of disease or agrochemical poisoning, and reported the changes in microbial communities in terms of relative abundance.

Q: I really missed information regarding absolute abundance of the bacterial and fungal communities in those samples, which could have been achieved by performing quantitative PCR. Adding this information to the study would not only improve the confidence of the results (since changes in microbial relative abundance do not always correlate with changes in absolute abundance) but also increase the impact of the study. I strongly recommend performing qPCR assays to investigate the overall bacterial and fungal loads in bee larvae and foragers.

A: Yes. We totally agree that the inclusion of qPCR results to reveal the overall microbial loads in our samples would definitely improve our current results.

However, the initial design of this study was to reveal the microbial variations between samples from different groups and isolate the causative pathogenic microbes in the cases of American Foulbrood (AFB) and chalkbrood disease (CBD). We have used up the spare AFB- and CBD-infected larvae samples for the isolation and identification of *P. larvae* and *A. apis*.

After receiving the comments of reviewers, we immediately contacted the company (Novogene Biological Information Technology Co., Ltd., Beijing, China) and asked them to return the remaining DNA samples to us under the project number of X101SC19050863-Z01-F016 (ITS1) and F015 (16S). We had planned to perform qPCR immediately using the readily available universal bacterial 16S rRNA gene primer pair of 27F and 355R in our lab. However, we were informed that there was almost no DNA samples left after the amplicon sequencing had been finished.

The situation now is that we have run out of DNA samples and field-samples, which denied us any possibility to perform this experiment. We feel sad and regretful. The lack of comprehensive consideration in the original experimental design put us in this helpless state now. The only thing we now can do is to learn from this lesson and save as much backup samples as possible in further studies.

Q: Another thing that caught my attention: more information should be provided regarding the potential agrochemicals used in the area, maybe the sampled bees could be assayed to detect the presence of potential agrochemicals. It is really vague to say that those bees were poisoned without proving that. As the authors mention, many other environmental stressors could be responsible for the observed side effects.

A: Yes. The analysis of residual agrochemicals on the surface and in the body of foragers would have been perfect for the specific identification of the source if we had backup forager samples.

In fact, the suspected poisoning of foragers by potential agrochemicals was largely dependent on our field investigation. After we had finished the sampling of the disordered foragers, we and the owner of the apiary drove around the nearby area for a field investigation. Besides the large area of oilseed rape field, we only found several very small pieces of vegetable fields scattered along the roadside which were almost 4 kilometers away from the apiary. Some peasants used these lands for the supply of extra vegetable for their family. We didn't find the exact owners of these vegetables. However, two passing peasants that we encountered told us pesticides, such as chlorpyrifos, emamectin benzoate, and thiamethoxam, were commonly used in this area for pest control at the early stage of vegetable growth. We suspected that the chemicals these peasants sprayed on the vegetables might drift with the wind to the closely neighbored oilseed rape field and be accidentally ingested by some foragers. That could explain why only foragers from several colonies were affected and exhibited abnormal symptoms.

As we mentioned in the manuscript, we eventually got our conclusion based on three points (1) no history of viral diseases outbreak in the apiary; (2) no prevalence of pathogenic bacteria or fungi detected; (3) death of foragers stopped when beekeepers transferred the apiary to another location 13 km away.

Other comments:

Q: Page 8, Line 25: Briefly explain how foragers were detected and collected. For the affected hives, were the sampled foragers exhibiting signs of poisoning? Were they collected at the entrance of the hive, leaving or returning to the hives?

A: We have added detailed information into the text, which are marked in blue.

Disordered foragers exhibited symptoms suspected of pesticide poisoning ~~presented with~~ which included unnaturally quick movements on the ground and lack of vitality after their return flight home at around 15:45-16:30 pm. Most of them died in close proximity to hives' entrances. For sample collection, foragers were collected at the entrance of the hive when they returned to the hives in the afternoon. Three disordered foragers per colony were collected and pooled, which were designated as group DIS. At the same time, 3 foragers per colony from colonies exhibiting no abnormal symptoms within the same apiary were collected, which were designated as group CT. DIS.

Q: Page 10, Line 48: Although the OTU method used in the study is completely valid, the new ESV/ASV methods are better in my opinion and should be considered for future work.

A: Thank you for your advice. Yes. We agree that ESV/ASV approaches are able to provide a more precise identification of microbes and a more detailed picture of the diversity within a sample. In our future work addressing the study of microbial compositions and variations we will consider to use this method.

Q: Figure 1 – The high proportion of “Others” is at least intriguing and deserves discussion.

A: “Others” referred to those microbes that came after the top 10, which were predominantly low-abundance microbes. We have added relevant discussions into the manuscript.

First part

In the present study, significant differences in microbial compositions were observed between larvae from ~~diseased and~~ healthy and diseased colonies. In healthy larvae, the top 10 bacterial and fungal genera accounted for 58.29% and 22.92% of the whole microbial composition in group CT.AFB and group CT.CBD respectively. The rest was mainly composed of low-abundance (relative abundance < 0.1%) microorganisms, which were actually the main part that contributed to the community diversity. In infected larvae, the pathogenic *P. larvae* and *A. apis* comprised the overwhelming majority of the bacterial and fungal microbiota in larvae from group AFB (96.48%) and group CBD (99.89%) respectively. The abundance of other microorganisms was too low to be detected. As a result, the community richness and diversity were significantly reduced in diseased larvae.

Second part

In the present study, six core gut taxa were all identified in foragers from both group CT.DIS and group DIS, which, altogether, accounted for 62.03% and 75.01% of the whole bacterial communities respectively. In previous reports, the proportion of these core bacterial members in the gut of adult honey bees accounted for over 95% of the community [55, 56]. We attributed this difference to different sampling methods. In this study, the whole-body microbiota of foragers was analyzed, not only the intestinal flora, but also the microorganisms in other parts of honey bees (such as the mouth parts and the hypopharyngeal glands), the inclusion of which may reduce the proportion of the top 10 microbes.

Q: Increase font size for Figures 1 and 3. It’s really hard to read them.

A: Yes. We have increased the font size in these figures.

Q: Page 3, Line 51: Remove the expression: “In the mean time”

A: We have deleted the phrase.

~~In the meantime~~, Significantly higher frequency of environmentally derived fungi was observed in disordered foraging bees, which reflected the perturbed microbiota communities of hosts.

Q: Page 4, Line 38: Replace “agent” with “agents”

A: We have replaced it.

The spore-forming bacterium *Paenibacillus larvae* (*P. larvae*) and the fungus *Ascosphaera apis* (*A. apis*) are the causative agents for AFB and CBD, respectively.

Q: Page 5, Line 25: Correct the expression: “are more tend to be”

A: We have changed the word “tend” to “inclined”.

For migratory apiaries, the microbiota of honey bees are more ~~tend~~ inclined to be influenced by exposure to the new environment

Q: Page 6, Line 27: Replace “microbiom” with “microbiome”

A: We have replaced it.

(2) the compositional and structural shifts taking place in the microbiome of foraging bees suspected of being exposed to agrochemicals in a third migratory apiary.

Q: Page 13, Line 35: “Similarly” followed by “However” in Line 40 is controversial

A: We deleted the sentence and kept the p value in the brackets

Similarly, results of the 16S rRNA amplicons sequencing showed that more OTUs were detected in group DIS (474) than that in group CT.DIS (358). ~~However, the difference was not statistically significant~~ ($p = 0.342$).

Q: Page 27, Line 49: Replace “recourses” with “resources”

A: We have replaced it.

We proposed that the lack of varied floral ~~recourses~~ resources throughout this season, together with reduction of flower

The above is our point-to-point response to reviewers and editors. All the other changes which had been suggested by the proofreader have been marked in the main text.

Thanks again for your valuable advice and efforts to improve the quality of our manuscript.

Best regards

Bin Zhou